# *Limosilactobacillus fermentum* 3872 That Produces Class III Bacteriocin Forms Co-Aggregates with the Antibiotic-Resistant *Staphylococcus aureus* Strains and Induces Their Lethal Damage

**DOI:** 10.3390/antibiotics12030471

**Published:** 2023-02-26

**Authors:** Vyacheslav M. Abramov, Igor V. Kosarev, Andrey V. Machulin, Tatiana V. Priputnevich, Evgenia I. Deryusheva, Ekaterina L. Nemashkalova, Irina O. Chikileva, Tatiana N. Abashina, Alexander N. Panin, Vyacheslav G. Melnikov, Nataliya E. Suzina, Ilia N. Nikonov, Marina V. Selina, Valentin S. Khlebnikov, Vadim K. Sakulin, Vladimir A. Samoilenko, Alexey B. Gordeev, Gennady T. Sukhikh, Vladimir N. Uversky, Andrey V. Karlyshev

**Affiliations:** 1Federal Service for Veterinary and Phytosanitary Surveillance (Rosselkhoznadzor) Federal State Budgetary Institution “The Russian State Center for Animal Feed and Drug Standardization and Quality” (FGBU VGNKI), 123022 Moscow, Russia; 2Kulakov National Medical Research Center for Obstetrics, Gynecology and Perinatology, Ministry of Health, 117997 Moscow, Russia; 3Skryabin Institute of Biochemistry and Physiology of Microorganisms, Federal Research Center “Pushchino Scientific Center for Biological Research of Russian Academy of Science”, Russian Academy of Science, 142290 Pushchino, Russia; 4Institute for Biological Instrumentation, Federal Research Center “Pushchino Scientific Center for Biological Research of Russian Academy of Science”, Russian Academy of Science, 142290 Pushchino, Russia; 5Laboratory of Cell Immunity, Blokhin National Research Center of Oncology, Ministry of Health RF, 115478 Moscow, Russia; 6Gabrichevsky Research Institute for Epidemiology and Microbiology, 125212 Moscow, Russia; 7Federal State Educational Institution of Higher Professional Education, Moscow State Academy of Veterinary Medicine and Biotechnology named after K.I. Skryabin, 109472 Moscow, Russia; 8Institute of Immunological Engineering, 142380 Lyubuchany, Russia; 9Department of Molecular Medicine, Morsani College of Medicine, University of South Florida, Tampa, FL 33612, USA; 10Department of Biomolecular Sciences, Faculty of Health, Science, Social Care and Education, Kingston University London, Kingston upon Thames KT1 2EE, UK

**Keywords:** *L. fermentum*, bacteriocin, antibacterial activity, *S. aureus*, antibiotic resistance

## Abstract

LF3872 was isolated from the milk of a healthy lactating and breastfeeding woman. Earlier, the genome of LF3872 was sequenced, and a gene encoding unique bacteriocin was discovered. We have shown here that the LF3872 strain produces a novel thermolabile class III bacteriolysin (BLF3872), exhibiting antimicrobial activity against antibiotic-resistant *Staphylococcus aureus* strains. Sequence analysis revealed the two-domain structural (lysozyme-like domain and peptidase M23 domain) organization of BLF3872. At least 25% residues of this protein are expected to be intrinsically disordered. Furthermore, BLF3872 is predicted to have a very high liquid-liquid phase separation. According to the electron microscopy data, the bacterial cells of LF3872 strain form co-aggregates with the *S. aureus* 8325-4 bacterial cells. LF3872 produced bacteriolysin BLF3872 that lyses the cells of the *S. aureus* 8325-4 mastitis-inducing strain. The sensitivity of the antibiotic-resistant *S. aureus* collection strains and freshly isolated antibiotic-resistant strains was tested using samples from women with lactation mastitis; the human nasopharynx and oral cavity; the oropharynx of pigs; and the cows with a diagnosis of clinical mastitis sensitive to the lytic action of the LF3872 strain producing BLF3872. The co-cultivation of LF3872 strain with various antibiotic-resistant *S. aureus* strains for 24 h reduced the level of living cells of these pathogens by six log. The LF3872 strain was found to be able to co-aggregate with all studied *S. aureus* strains. The cell-free culture supernatant of LF3872 (CSLF3872) induced *S. aureus* cell damage and ATP leakage. The effectiveness of the bacteriolytic action of LF3872 strain did not depend on the origin of the *S. aureus* strains. The results reported here are important for the creation of new effective drugs against antibiotic-resistant strains of *S. aureus* circulating in humans and animals.

## 1. Introduction

Staphylococci are typical representatives of the normal microbiota, colonizing the skin and mucous membranes of healthy people and farm animals, while the frequency of colonization by *Staphylococcus aureus* of open biocenoses is 25–30% [1,2]. At the same time, *S. aureus* could be widespread pathogens that cause various infections in humans and animals, ranging from minor skin lesions to life-threatening conditions, such as meningitis, endocarditis, and pneumonia [2,3,4]. Antibiotics containing the β-lactam ring (natural and synthetic penicillins, cephalosporins of I, II, III, and IV generations, carbapenems, and monobactams) are widely used for the treatment of infectious diseases caused by *S. aureus* [5,6,7]. Penicillins, cephalosporins, and monobactams are sensitive to the hydrolyzing action of special β-lactamase enzymes produced by various pathogens, including *S. aureus* strains [8]. To overcome the β-lactamase resistance of pathogens, protected penicillins (clavulanic acid, sulbactam, and tazobactam) were developed. The bactericidal effect of β-lactam antibiotics is associated with the selective inhibition of various stages of the construction of peptidoglycan of the cell wall of pathogenic microorganisms, including *S. aureus*. The destruction of cell wall synthesis under the action of β-lactams causes inhibition of *S. aureus* cell growth [9]. The growth of *S. aureus* resistant to antimicrobial drugs is currently one of the most acute global public health problems, leading to the failure in the treatment of infectious endocarditis, meningitis, osteomyelitis, and septic arthritis [10]. *S. aureus*, being considered one of the most significant etiological factors of infection, developed multiple mechanisms of antibiotic resistance, which are transferred rapidly between strains in both hospital and community settings [10,11,12].

In the case of methicillin-resistant *S. aureus* (MRSA), the problem is particularly evident: the MRSA was previously predominant in hospital settings as hospital-acquired MRSA (HA-MRSA), but now, it is being detected in community settings as community-acquired MRSA (CA-MRSA), showing high infectivity and virulence [13,14,15,16,17,18]. CA-MRSA strains differ from HA-MRSA strains: they have an extra genome, carry different elements of the staphylococcal cassette chromosome mec (SCCmec), affect different populations, and cause clinical symptoms that differ from those of the HA-MRSA infection [13,14,15,19,20,21,22,23,24,25,26,27,28].

The increase in the number of multi-drug resistant *S. aureus* strains and the shortage of new antibiotics on the market are becoming a serious public health problem. Therefore, experts of the World Health Organization (WHO) have included MRSA in the list of microorganisms that require the urgent creation of new antibiotics to combat these pathogens [29]. 

The ineffectiveness of previously used antibiotic therapies warrants research on alternative treatments, such as phage therapy [30,31,32] and probiotic therapy [33,34]. The sensitivity of eradicated bacteria to bacteriophages is one of the prerequisites of successful phage therapy. However, similar to antibiotic resistance, bacteria can also develop resistance to bacteriophages [32].

Strains of lactic acid bacteria isolated from the breast milk of healthy women have shown a high efficiency as oral probiotics for the treatment of such conditions [35,36]. *Lactiplantibacillus plantarum* ATCC 8014 strain and its antimicrobial compounds showed great potential to inhibit the growth of *S. aureus* isolated from bovine mastitis [37]. However, the molecular mechanisms of the antibacterial effects of lactic acid bacteria against pathogenic staphylococci have not been sufficiently studied. Earlier, we showed that co-cultivation of *S. aureus* 8325-4 cells with the LF3872 cells for 24 h reduced the level of living test culture by six log and that such an effect was alleviated by heat treatment of LF3872 [38]. The *S. aureus* 8325-4 strain is used to simulate lactation mastitis in laboratory animals [39]. 

In a previous study [40], analysis of the LF3872 genome discovered a gene encoding a class III bacteriocin BLF3872. Bacteriocins have antimicrobial activity against pathogenic microorganisms, which determines their biotechnological potential [41,42]. Bacteriocins inhibit the growth of target microorganisms: they affect the cell wall and influence the gene expression and protein production within cells [43]. Class III bacteriocins are antimicrobial thermos-sensitive proteins with a molecular weight of more than 30 kDa that are capable of degrading cell wall murein [44]. This group includes some colicins, zoocins, megacins (*Bacillus megaterium*), klebicin (*Klebsiella pneumonia*), helveticins I and J (*Lactobacillus helveticus*), and enterolysin A (*Enterococcus faecalis*) [45]. The new class III bacteriocin BLF3872 from LF3872 has not yet been studied.

Here, we analyze the primary structure of a new class III bacteriocin BLF3872 and model its spatial organization. Moreover, we studied the changes in the cellular morphology of *S. aureus* 8325-4 cells after the co-cultivation with LF3872 cells using scanning electron microscopy (SEM), and the co-aggregative activity, as well as the antibacterial properties of LF3872 against antibiotic-resistant collection strains and freshly isolated strains in samples from women with lactation mastitis; from the human nasopharynx and oral cavity; from the oropharynx of pigs; and from cows with a diagnosis of clinical mastitis.

## 2. Results and Discussion

### 2.1. A Novel Class III Bacteriocin BLF3872

The predicted size and molecular weight of the identified BLF3872 corresponds to a protein with a length of 404 residues, a molecular weight of approximately 42.5 kDa, and an isoelectric point of 5.15. The protein contains two cysteine residues and is classified by using ProtParam as stable (resistance index is 36.0). As was shown earlier [38], the 280–404 region of the BLF3872 corresponds to the peptidase M23 domain (residues 279–377; PF01551). Furthermore, the N-terminal half of this protein includes a lysozyme-like domain (residues 70–199; PF18013). Sequence comparison of the BLF3872 with the class III bacteriocins revealed the conservative motif (e.g., His280 and Asp284 in the HXXXD motif, and His361 and His363 in the HXH motif) potentially capable of coordinating the Zn^2+^ ion.

Figure 1A shows the 3D structure of BLF3872 modeled by using AlphaFold and shows that this protein has two spatially well-separated N- and C-terminal domains (residues 70–199 and 279–377, respectively), with a long N-terminal unordered region (residues 1–60) and a flexible linker between domains (Figure 1A). In line with the earlier expectations [38], the structures of the N- and C-domains resemble lysozyme-like enzymes and metalloendopeptidases (peptidase M23 domain), respectively.

The BLAST sequence search identified the closest similarity of the BLF3872 to Morphogenesis protein 1 (365 a.a.) from *Bacillus* phage phi29 (UniProt ID P15132). Figure 1B shows a 3D X-ray crystal structure of the Morphogenesis protein 1 and shows that BLF3872 and Morphogenesis protein 1 are structurally rather similar despite a low level of the amino acid sequence identity (~30% according to EMBOSS Needle). Note that the amino acid sequence of the BLF3872 protein is depicted in Figure 2A. Based on these findings, one can hypothesizes that BLF3872 is capable of a cleavage of both peptide cross-links and the polysaccharide backbone of the peptidoglycan cell, thereby acting as a peptidoglycan-degrading enzyme [47].

According to AlphaFold, as BLF3872 is expected to have long flexible regions, we evaluated the intrinsic disorder predisposition of this protein using a set of commonly utilized disorder predictors. The results of these analyses are summarized in Figure 2B, which clearly shows that at least 25% residues of this protein are expected to be disordered, with most disordered regions being located outside the lysozyme and peptidase M23 domains.

Curiously, the N-terminal lysozyme-like domain is expected to be more flexible than the C-terminal peptidase M23 domain. Since intrinsically disordered proteins are commonly involved in liquid-liquid phase separation (LLPS) and related biogenesis of various membrane-less organelles (MLOs) or biomolecular condensates [50,51,52,53,54,55,56,57], at the next stage, we checked the predisposition of BLF3872 for phase separation. Figure 2C shows that this bacteriocin is characterized by a very high LLPS propensity of 0.9887 and contains six DPRs (residues 31–72, 82–102, 106–134, 204–246, 253–281, and 392–494). Therefore, it is likely that BLF3872 can phase separate by itself. This conclusion is further supported by the results of the PSPredictor [58] analysis, according to which this protein has a very high PSP score of 0.9955 as well. This is an interesting observation suggesting that at least some of the functions of the bacteriocin BLF3872 can be associated with the formation of biomolecular condensates. 

In addition to the evaluation of the LLPS predisposition of a query protein, FuzDrop is capable of finding aggregation hot-spots, which are defined as residues/regions that may promote the conversion of the liquid-like condensed state into a solid-like amyloid state [59,60]. Figure 2C shows that the bacteriocin BLF3872 contains seven such aggregation hot-spots (residues 34–43, 81–87, 117–123, 206–211, 265–270, 274–281, and 292–297). Furthermore, FuzDrop can evaluate the probability of a protein to have residues/regions with cellular context dependence that are capable of switching between different binding modes depending on the peculiarities of cellular environment [49]. Figure 2C demonstrates that BLF3872 contains 17 regions with the context-dependent interactions (residues 15–21, 34–43, 80–87, 99–109, 117–123, 130–136, 147–154, 193–199206-211, 242–248, 252–257, 265–270, 274–281, 285–300, 319–325, 363–371, and 380–397). Importantly, these regions overlap with, are included into, or are located in close proximity to the disordered or flexible regions found in BLF3872. This is in line with known notion that the heterogeneity or disorder of the bound structure can vary with cellular conditions [49].

### 2.2. Scanning Electron Microscopy

The results of antibacterial action of LF3872 strain producing a novel class III bacteriocin (bacteriolysin) BLF3872 on *S. aureus* 8325-4 strain are shown in Figure 3. The cell wall of intact *S. aureus* 8325-4 strain is a homogeneous electron-dense layer (Figure 3A). It is known that peptidoglycan (PG) is a key component of *S. aureus* cell wall that helps maintain cell shape and prevents bacteria from lysis through osmotic rupture [61]. PG is composed of polysaccharide chains cross-linked by short peptides. The polysaccharide chains contain N-acetylmuramic acid (NAM) and N-acetylglucosamide (NAG). In addition to these basic polymers, the cell walls of *S. aureus* contain small amounts of lipids, polysaccharides, and proteins. The initial stage of the antibacterial action of LF3872 cells is their co-aggregation with *S. aureus* 8325-4 cells (Figure 3B–D). The genome of LF3872 contains a gene, the product of which is responsible for the co-aggregation of the cells of this strain with pathogens [40,62]. Lysozyme (or muramidase or N-acetylmuramic acid hydrolase, EC 3.2.1.17) is known to be a protein with the enzymatic activity through the hydrolysis of β-1,4-glycosidic bonds between NAM and NAG in the PG of *S. aureus* bacterial cell wall [63,64].

We assume that the lysozyme-like N-terminal domain of BLF3872 hydrolyzes the β-1,4-glycosidic bonds between NAM and NAG in PG of *S. aureus* 8325-4 bacterial cell wall and metalloendopeptidase-like C-domain of BLF3872 hydrolyzes amide bonds between amino acids of short peptides connecting polysaccharide chains of PG. The BLF3872 destroys the PG of the cell wall of *S. aureus* 8325-4 strain and induces the cell death (Figure 3B–D). BLF3872 has a homology with phage lysins, in which one domain has the muramidase activity and the second domain has the amidase activity. These domains exhibit synergistic antibacterial action on the pathogens and, at the same time, practically do not affect commensal bacteria [65]. Thus, BLF3872 secreted by the LF3872 strain reaches the PG layer of *S. aureus* 8325-4 cells, destroys it, and causes cell death.

### 2.3. Antibacterial Activity of LF3872 against Collection ATCC Methicillin-Resistant S. aureus (MRSA) Strains

The antagonistic activity of LF3872 against a collection of the ATCC methicillin-resistant *S. aureus* (MRSA) strains is shown in Table 1. In the co-culture experiments, MRSA strains ATCC BAA-1683, ATCC BAA-2313, ATCC 33592, and ATCC BAA-2094 were highly sensitive to the bacteriolytic action of LF3872. The co-cultivation of ATCC MRSA strains with the LF3872 for 24 h reduced the level of live test cultures by six log, while the heat-treated LF3872 has no effect on the level of live test cultures of ATCC MRSA strains.

WHO experts have included MRSA in the list of pathogens that require the urgent creation of new effective antibiotics [29]. BLF3872 effectively destroys MRSA strains ATCC BAA-1683, ATCC BAA-2313, ATCC 33592, and ATCC BAA-2094. It is known that *S. aureus* strains stored in the collection may be less resistant to the antibacterial drugs [66]. To address this point, in the following experiments, we used clinical strains directly isolated from humans and farm animals.

### 2.4. Antibacterial Activity of LF3872 against S. aureus Clinical Isolates from the Milk of Women with Mastitis 

The antagonistic activity of LF3872 against *S. aureus* clinical isolates from the milk of women with mastitis is shown in Table 2. In the co-culture experiments, antibiotic-resistant *S. aureus* IIE CI-SA Hu 1246 strain, *S. aureus* IIE CI-SA Hu 1247 strain, *S. aureus* IIE CI-SA Hu 1248 strain, and *S. aureus* IIE CI-SA Hu 1249 strain were highly sensitive to the bacteriolytic action of LF3872. The co-cultivation of *S. aureus* clinical isolates from the milk of women with mastitis with the LF3872 for 24 h reduced the level of living test cultures by six log. However, the heat-treated LF3872 did not reduce the level of live *S. aureus* clinical isolates from the milk of women with mastitis. 

*S. aureus* is the main etiological agent of acute mastitis [67,68] but more frequently causes subclinical infections that tend to become chronic and difficult to eradicate with conventional anti-microbial therapies [69]. The vertical transmission of multidrug-resistant *S. aureus* from mother to child has been established [70]. LF3872 is promising for the prevention of *S. aureus*-induced lactation mastitis in women during breast-feeding of a child and can be used to implement a strategy for preventive control of this widespread disease in the world. 

### 2.5. Antibacterial Activity of LF3872 against S. aureus Strains Isolated from the Nasopharynx and Oral Cavity of Humans

The antagonistic activity of LF3872 against *S. aureus* strains isolated from the nasopharynx and oral cavity of humans is shown in Table 3. In the co-culture experiments, the antibiotic-resistant *S. aureus* IIE CI-SA Hu 1261 strain, S. *aureus* IIE CI-SA Hu 1262 strain, *S. aureus* IIE CI-SA Hu 1263 strain, and *S. aureus* IIE CI-SA Hu 1264 strain isolated from the nasopharynx and oral cavity of humans were highly sensitive to the bacteriolytic action of LF3872. The co-cultivation of *S. aureus* strains isolated from the human nasopharynx and oral cavity with LF3872 for 24 h reduced the level of living test cultures by six log. Heat-treated LF3872 has no effect on the level of live *S. aureus* strains isolated from the nasopharynx and oral cavity of humans. 

Despite growing knowledge of the oral microbiome, the roles of *S. aureus* in oral diseases and the risk of spreading staphylococcal infection from this reservoir are still not fully understood [71]. The prevalence of *S. aureus* infections has increased recently. The burden of infection due to the MRSA strains is significantly more evident in children than in other age groups [72]. *S. aureus* colonization of the nasopharynx begins very early, in the first six months of life [73]. The use of the LF3872 strain as a probiotic may help in the prevention and eradication of antibiotic-resistant *S. aureus* in the nasopharynx and oral cavity in humans.

### 2.6. Antibacterial Activity of LF3872 against S. aureus Strains Isolated from the Oropharynx of Pigs 

The antagonistic activity of LF3872 against *S. aureus* strains isolated from the oropharynx of pigs is shown in Table 4. In the co-culture experiments, the antibiotic-resistant *S. aureus* IIE CI-SA Pi 1345 strain, *S. aureus* IIE CI-SA Pi 1347 strain, *S. aureus* IIE CI-SA Pi 1356 strain, and *S. aureus* IIE CI-SA Pi 1357 strain isolated from the oropharynx of pigs proved to be highly sensitive to the bacteriolytic action of LF3872. The co-cultivation of *S. aureus* strains isolated from the oropharynx of pigs with LF3872 for 24 h reduced the level of living test cultures by six log. Heat-treated LF3872 had no effect on the level of live *S. aureus* strains isolated from the oropharynx of pigs. 

Pigs are the main source of the spread of antibiotic-resistant strains of *S. aureus.* Numerous studies have shown that people living or working on pig farms, including farmers and their families, veterinarians and slaughterhouse workers, are at increased risk of colonization or infection with the LA-MRSA [74,75,76,77,78,79]. Direct contact is the main route of transmission of MRSA between pigs [80,81]. It has been suggested that, in the process of the transmission of these bacteria to negative animals, the MRSA-positive pigs may play a critical role (horizontal transmission) [80,81]. Only a few positive animals can lead to the spread of MRSA on or even off farms [82,83]. It is also possible that MRSA can be transmitted from sows to their offspring (vertical perinatal transmission) [80,83,84,85,86]. The use of a feed additive containing the LF3872 strain may help in the prevention and eradication of the MRSA on pig farms.

### 2.7. Antibacterial Activity of LF3872 against S. aureus Clinical Isolates from the Milk of Cows with Mastitis

The antagonistic activity of LF3872 against *S. aureus* clinical isolates from the milk of cows with mastitis is shown in Table 5. In the co-culture experiments, the antibiotic-resistant *S*. *aureus* IIE CI-SA Co 1280 strain, *S. aureus* IIE CI-SA Co 1281 strain, *S. aureus* IIE CI-SA Co 1282 strain, and *S. aureus* IIE CI-SA Co 1283 strain were highly sensitive to the bacteriolytic action of LF3872. The co-cultivation of clinical isolates of *S. aureus* from the milk of cows with mastitis, with LF3872 for 24 h, reduced the level of live test cultures by six logs. Heat-treated LF3872 had no effect on the level of live clinical isolates of *S. aureus* from the milk of cows with mastitis. The WHO One Health emphasizes the importance of considering human and animal health in a collaborative and interrelated manner [87].

The use of a feed additive containing the LF3872 strain may help in the prevention and treatment of lactation mastitis in cows. 

### 2.8. Cells of LF3872 Co-Aggregate with Antibiotic-Resistant S. aureus Strains Isolated from Humans and Animals

We evaluated the co-aggregation capacities of LF3872 with antibiotic-resistant *S. aureus* strains isolated from humans and animals and ATCC MRSA strains. LF3872 is a strong co-aggregator of antibiotic-resistant *S. aureus* strains isolated from humans and animals and ATCC MRSA strains. 

At pH 7.0 in phosphate-buffered saline (PBS) buffer, LF3872 co-aggregated with all *S. aureus* strains analyzed in this study. The percentage of co-aggregation ranged from 45.9 ± 6.5% to 54.5 ± 4.7%. LF3872 also co-aggregated with all tested *S. aureus* strains at pH 5.0 in 2-(N-morpholino)ethanesulfonic acid (MES) buffer.

At lower pH values, the percentage of co-aggregation significantly increased (*p* < 0.05), ranging from 80.6 ± 5.8% to 89.3 ± 7.6% (Table 6). Co-aggregation increased with decreasing pH. Some lactobacilli use co-aggregation mechanisms to realize their probiotic properties [88,89,90,91]. The ability to co-aggregate with pathogens is one of the most important manifestations of the antagonistic activity of lactobacilli [92,93]. The formation of lactobacilli co-aggregates with pathogens leads to inhibition of the adhesion of these pathogens to the host epithelial cells [94,95]. Lactobacilli use co-aggregation mechanisms to create a hostile microenvironment around the pathogen [95]. They secrete antibacterial metabolites inside the co-aggregates, such as bacteriocins [94,96], organic acids, and H_2_O_2_ [97]. In various biotopes of the macro-organism, the co-aggregation process of lactobacilli with pathogens is important for maintaining a homeostatic microbiota [98]. Due to the increase of antibiotic resistance in pathogens, it is relevant to identify lactobacilli with high co-aggregation activity [95,99]. Thus, the co-aggregation ability of LF3872 may prevent the colonization of human and animal organs and tissues by antibiotic-resistant strains of *S. aureus*. 

### 2.9. CSLF3872 Induces Cell Damage and ATP Leakage in Antibiotic-Resistant S. aureus Strains 

Cell-free culture supernatant of LF3872 (CSLF3872) induced cell damage and ATP leakage in antibiotic-resistant *S. aureus* strains isolated from humans, animals, and in ATCC MRSA strains. ATP leakage levels were examined based on non-selective pore formation and cell damage indexes. The research results are presented in Table 7.

The cultivation of various antibiotic-resistant *S. aureus* strains in the presence of CSLF3872 for 2.5 h increased the level of extracellular ATP on average from 6.5 (nm/OD) to 346.0 (nm/OD) (*p* < 0.001). Since the LF3872 strain produces thermolabile class III bacteriocin BLF3872, heating of the CSLF3872 leads to the loss of its ability to damage the cell wall and ATP leakage in antibiotic-resistant *S. aureus* strains. The boiled CSLF3872 did not induce cell damage and ATP leakage in antibiotic-resistant *S. aureus* strains isolated from humans, animals, and in ATCC MRSA strains. In a previous study [100], the ability of the thermolabile recombinant class III bacteriocin Helveticin-M from *L. crispatus* to destroy the cell wall of *S. aureus* was demonstrated. Helveticin-M also disrupted the inner membrane, as confirmed by leakage of intracellular ASP from *S. aureus* cells. As CSLF3872 may additionally contain lactic acid and hydrogen peroxide, which in high concentrations are also able, along with BLF3872, to inhibit the growth of *S. aureus* and damage the inner membrane [101,102], we studied lactic acid and hydrogen peroxide production by LF3872. The results of the studies are given in (2.10; 2.11).

### 2.10. Lactic Acid Production by LF3872 Strain in the Process of Its Cultivation in the MRS Broth 

The results of the studies shown in Table 8 demonstrate that the LF3872 strain produces an insignificant amount of lactic acid during 20 h of observation (210 ± 8 µg/mL). In a previous study [103], the lytic effect of CS strain *L. fermentum* 1912 and strain *L. plantarum* 8513 on *S. aureus* cells was studied. The lytic effect of lactic acid and bacteriocins as separate components of CS strain *L. fermentum* 1912 and CS strain *L. plantarum* 8513 on *S. aureus* cells was also studied. Both strains of lactobacilli were super-producers of lactic acid. Strain *L. fermentum* 1912 increased the concentration of lactic acid in CS to 6.430 ± 0.063 mg/mL; strain *L. plantarum* 8513 increased the concentration of lactic acid in CS to 10.796 ± 0.074 mg/mL. It was found that bacteriocins from CS strain *L. fermentum* 1912 and strain *L. plantarum* 8513, but not lactic acid cause lysis of *S. aureus* cells [103]. According to [103], CS of LF3872 strain with such a low lactic acid level cannot cause *S. aureus* cell lysis. Therefore, it can be assumed that LF3872 strain lyses *S. aureus* cells (Figure 3A) due to the production of BLF3872.

### 2.11. Hydrogen Peroxide Production by LF3872 Strain in the Process of Its Cultivation in the MRS Broth 

The results of the studies shown in Table 9 demonstrate that the LF3872 strain produces an insignificant amount of H_2_O_2_ during 20 h of observation (0.038 mM/10^7^ CFU). In a previous study [104], the lytic effect of the CS of *L. gasseri* CRL 1421 strain, a superproducer of H_2_O_2_, and the CS of *L. gasseri* CRL 1412 strain, producing a small amount of H_2_O_2_, on *S. aureus* cells was studied. The mean amount of H_2_O_2_ generated by *L. gasseri* CRL 1421 was greater than that produced by LF3872 (1517.92 mM/10^7^ CFU vs. 0.038 mM/10^7^ CFU, *p* < 0.01). The CS of *L. gasseri* CRL 1421 strain induced intensive damage to the cell structure of *S. aureus*; disintegration of the cell wall observed using transmission electron microscopy [104]. At the same time, *L. gasseri* CRL 1412 strain producing a small amount of H_2_O_2_ (0.041 mM/10^7^ CFU) at the level of LF3872 strain did not cause damage to *S. aureus* cells. Based on the research results available in the literature [104], the CS of LF3872 strain with such a low H_2_O_2_ level cannot cause *S. aureus* cell lysis. Therefore, it can be assumed that LF3872 strain lyses *S. aureus* cells (Figure 3A) due to the production of BLF3872.

## 3. Materials and Methods

### 3.1. Used Microorganisms and Their Growth Conditions 

LF3872 [40,62,105,106] was deposited in the Collection of Microorganisms of the Institute of Immunological Engineering, Lyubuchany, Moscow region, Russia.

A full list of used microorganisms and their growth conditions are given in Table 10.

### 3.2. Sequence Alignment and Bioinformatic Analysis

ProtParam (https://web.expasy.org/protparam/; accessed on 2 January 2023) was used for sequence analysis [107]. A sequence similarity search was performed using the Basic Local Alignment Search Tool (BLAST) (http://www.ncbi.nlm.nih.gov/BLAST; accessed on 2 January 2023). The pairwise sequence alignment of BLF3872 and Morphogenesis protein 1 from Bacillus phage phi29 (UniProt ID: P15132) was implemented by the EMBOSS Needle (https://www.ebi.ac.uk/Tools/psa/emboss_needle/; accessed on 2 January 2023). The tertiary structure of Morphogenesis protein 1 from Bacillus phage phi29 (chain A of 3CSQ, X-RAY) was taken from the PDB bank (https://www.rcsb.org/; accessed on 2 January 2023). The tertiary structure of BLF3872 was modeled using AlphaFold (https://alphafold.ebi.ac.uk/; accessed on 2 January 2023) [46]. The tertiary structure models were drawn with molecular graphics system PyMOL v.1.8 (https://pymol.org/; accessed on 2 January 2023).

Intrinsic disorder predisposition was quantified on a per-amino acid residue basis using the Rapid Intrinsic Disorder Analysis Online (RIDAO) web server (https://ridao.app/; accessed on 12 January 2023) [48]. In these the analyses, values of 0 and 1 correspond to the most ordered and most disordered residues, respectively. 

The FuzDrop web server (https://fuzdrop.bio.unipd.it/predictor; accessed on 12 January 2023), which is capable of finding droplet promoting regions (DPRs), the aggregation-promoting regions (aggregation hot spots), and residues/regions with the cellular context dependence [49] in a protein, was utilized to evaluate the propensity of a query protein for phase separation [60,108,109]. The FuzDrop output shows an interactive graph showing distribution of the per single residue droplet-promoting probabilities (pDP) and gives an estimated value of the probability of spontaneous liquid-liquid phase separation (LLPS; p_LLPS_), which is used for protein classification as droplet drivers (p_LLPS_ ≥ 0.60), droplet-clients (p_LLPS_ < 0.60 and the presence of at least one DPR), or unrelated to LLPS (p_LLPS_ < 0.60 and no DPRs) [60,108,109]. The phase separation potential of a query protein was further validated using PSPredictor (http://www.pkumdl.cn:8000/PSPredictor/; accessed on 12 January 2023) [58].

For a query protein, PSPredictor generates a PSP score ranging from 0 to 1 in a tabulated form and shows the PSP status of a query protein in a “Yes/No” form. A query protein can be considered as a potential phase separating protein if its PSP score is ≥0.50.

### 3.3. Scanning Electron Microscopy

For experiments on SEM, intact *S. aureus* 8325-4 cells or co-aggregates of LF3872 cells with *S. aureus* 8325-4 cells were preliminarily fixed in a 2.5% solution of glutaraldehyde in the cacodylate buffer overnight at 4 °C. Additionally, after three washes in same buffer, the cells were fixed in 1% solution of OsO_4_ in cacodylate buffer at 20 °C for 3 h. After fixation and subsequent washing in buffer, the cells were dehydrated in increasing concentrations of ethanol (from 30 to 100%) for 20 min at each stage. Next, the cells were placed in tert-butanol and left at 4 °C for 12 h. The studied samples were freeze-dried in JFD-320 (JEOL, Tokyo, Japan) and were placed on a disk and sputtered with a gold in a JEE-4X (JEOL, Tokyo, Japan) vacuum evaporator. Images were taken on the JSM-6510LV (JEOL, Tokyo, Japan) microscope.

### 3.4. Determination of Antibacterial Activity of LF3872 against the Studied Groups of S. aureus Strains 

The antibacterial activity of the LF3872 strain against the studied groups of *S. aureus* strains was determined by using a previously described method [38]. Briefly, in 20 mL of the TGVC medium, 1 mL of the LF3872 inoculum grown on MRC medium and the test *S. aureus* strains were co-cultivated. After 24 h, the amount of different *S. aureus* cells (grown on TGVC medium in monoculture (control) and in the presence of lactobacilli) was counted. Aliquots were taken under aseptic conditions, serially diluted, and put on TGVC agar. After incubation, *S. aureus* strains colonies were counted and expressed as colony-forming units per milliliter (CFU/mL). Incubation was performed under aerobic conditions (24 h at 37 °C). 

### 3.5. Co-Aggregation Assay for the Determination of Interactions between LF3872 and S. aureus Strains 

Co-aggregation between LF3872 and *S. aureus* strains of the studied groups was determined by measuring the optical density (OD) at 600 nm, as previously described [38]. As pH can affect the co-aggregation, the MES buffer pH 5 (Sigma-Aldrich, St. Louis, MO, USA) for cell suspension was used instead of PBS.

### 3.6. Preparation of Cell-Free Culture Supernatant (CSLF3872)

CSLF3872 was prepared as previously described [38]. The lyophilized sediment of CSLF3872 was used to study cytoplasmic membrane permeability in *S. aureus* cells of various strains. Briefly, under anaerobic conditions at 37 °C, LF3872 was grown in MRS broth. Then, the culture was diluted in MRS broth and was grown anaerobically for two days. CSLF3872 was sequentially collected via centrifugation, and then filter-sterilized and concentrated via speed-vacuum drying.

### 3.7. Assessment of Cytoplasmic Membrane Permeability by Measurement of Extracellular ATP in S. aureus Strains of the Studied Groups 

Measurement of extracellular ATP in *S. aureus* strains of the studied groups for the assessment of cytoplasmic membrane permeability was performed as previously described [38]. Briefly, *S. aureus* strains were centrifuged and resuspended (PBS (pH 7.0), OD_600_ = 1.0 (5 × 10^8^ CFU/mL)). Then, bacteria were processed in the presence of CSLF3872 (for 2.5 h at 37 °C). After CSLF3872 treatment, the ATP detection kit (Beyotime, Shanghai, China) was used for the detection of extracellular ATP levels. For luminescence detection, an Infinite 200 PRO microplate reader (Tecan, Männedorf, Switzerland) was used. 

### 3.8. Lactic Acid Determination

The suspension of LF3872 cells grown in the MRS broth was centrifuged at 5000× *g* for 20 min at 4 °C. The culture supernatant was filtered using a 0.22 µm pore size filter (Millipore, Burlington, MA, USA).

The concentration of lactic acid in CSLF3872 produced by the LF3872 strain was determined according to the method [110] using a high performance liquid chromatography (HPLC) system equipped with an ultraviolet-visible detector (Shidmadzu, Kyoto, Japan) set at 220 nm and a Luna C18(2) column (150 mm × 4.6 mm, 5 µm; Phenomenex, Torrance, CA, USA). HPLC grade lactic acid (Sigma-Aldrich, St. Louis, MO, USA) was used as the standard.

### 3.9. Hydrogen Peroxide Determination

The suspension of LF3872 cells grown in the MRS broth was centrifuged at 5000× *g* for 20 min at 4 °C. The culture supernatant was filtered using a 0.22 µm pore size filter (Millipore, Burlington, MA, USA). The concentration of hydrogen peroxide was determined as described previously [101], calorimetrically through the detection of the absorbance induced with chromophore: 3,3′,5,5′-tetramethylbenzidine (TMB) in the presence of a horseradish peroxidase (Sigma-Aldrich, St. Louis, MO, USA).

### 3.10. Statistical Analysis

The results were analyzed using one-way analysis of variance (ANOVA) and represented as the Mean ± SD of six independent experiments, tested in triplicate. Statistical significance was evaluated by using the Student’s *t*-tests. Results were considered significant at *p* < 0.05.

## 4. Conclusions

The LF3872 strain has the potential of producing a novel class III bacteriocin BLF3872 with two predicted domains: an N-terminal domain with a lysozyme-like structure and a C-terminal domain with a metalloendopeptidase-like structure. BLF3872 is predicted to have noticeable levels of intrinsic disorder and is expected to serve as an LLPS driver, suggesting that at least some functions of this protein are related to the formation of biomolecular condensates. LF3872 is shown to form co-aggregates with both collection antibiotic-resistant *S. aureus* strains and with *S. aureus* strains freshly isolated from humans and animals. The co-aggregation of the LF3872 strain with various *S. aureus* strains is the initial stage of its antibacterial activity. According to electron microscopy data, LF3872 producing bacteriocin BLF3872 lyses the cells of *S. aureus* 8325-4 strain. Co-cultivation of the LF3872 strain with various antibiotic-resistant *S. aureus* strains for 24 h reduced the level of living cells of this pathogen by six log. The cell-free culture supernatant of LF3872 (CSLF3872) induced *S. aureus* cell damage and ATP leakage. Probiotic drugs and feed additives created in future on the basis of the LF3872 strain may be promising for the prevention of lactation mastitis induced by *S. aureus* in women during breastfeeding and in cows during the lactation period.

## Figures and Tables

**Figure 1 antibiotics-12-00471-f001:**
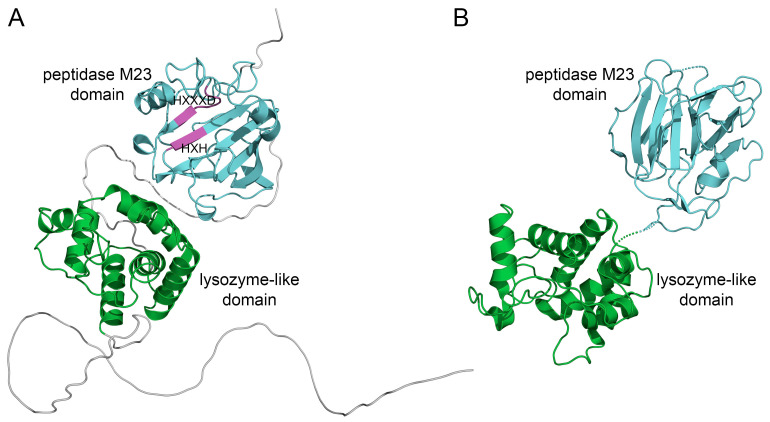
(**A**) Model structure of BLF3872 obtained by using AlphaFold [46]. The N- and C-terminal domains are highlighted using green and cyan, respectively. Conservative motifs HXXXD and HXH in the peptidase M23 domain are marked using magenta. (**B**) The 3D structure of Zn^2+^ loaded Morphogenesis protein 1 from *Bacillus* phage phi29 (chain A of 3CSQ, X-RAY). The lysozyme-like domain and peptidase M23 domain are highlighted using green and cyan, respectively.

**Figure 2 antibiotics-12-00471-f002:**
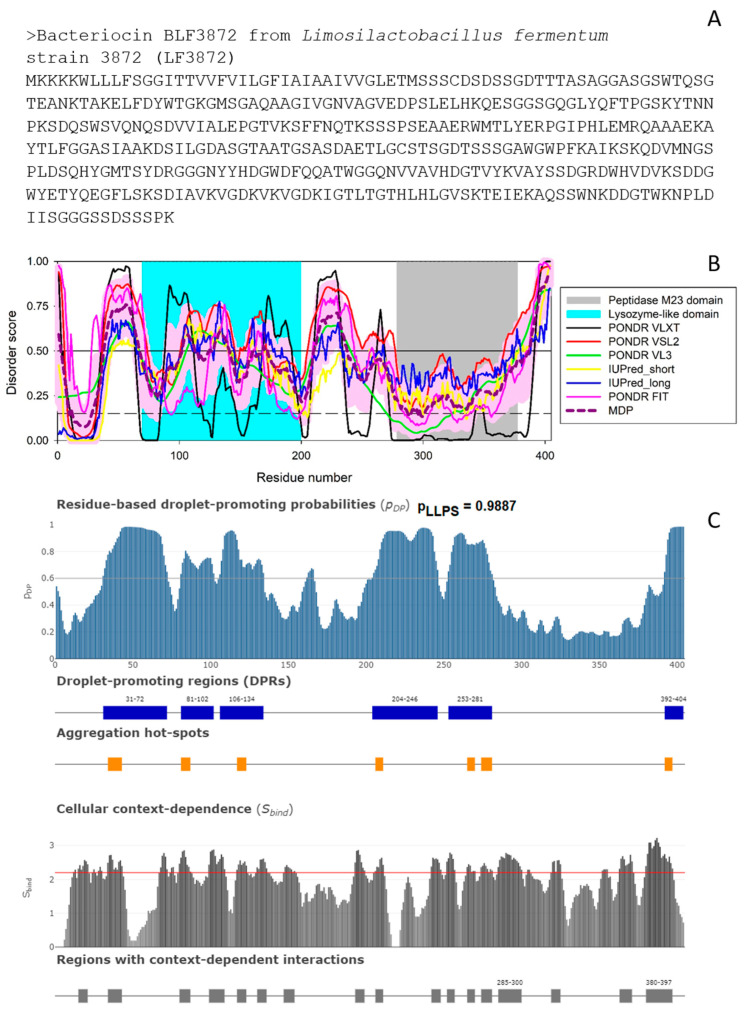
(**A**). Amino acid sequence of the bacteriocin BLF3872. (**B**). Intrinsic disorder predisposition evaluated using RIDAO [48] that aggregates the results from a number of well-known disorder predictors: PONDR^®^ VLXT (black line), PONDR^®^ VLS2 (red line), PONDR^®^ VL3 (green line), PONDR^®^ FIT (pink line), IUPred2_long (blue line), and IUPred2_short (yellow line). A thick dashed dark-pink line represents the mean disorder prediction (MDP) profile calculated by averaging the outputs of these six predictors, whereas a light pink shadow shows the corresponding error distribution. The results of residual disorder propensity estimation using these tools are presented as real numbers from 0 (ideal order prediction) to 1 (ideal disorder prediction). To identify disordered residues and regions in the proteins of interest, a threshold of ≥0.5 was used. Solid and dashed horizontal lines at disorder scores 0.5 and 0.15 correspond to the disorder and flexibility thresholds. (**C**). Evaluation of the LLPS propensity of the bacteriocin BLF3872 using FuzDrop. The top plot represents the sequence distribution of the droplet- or LLPS-promoting probability (p_LLPS_) and shows that BLF3872 is characterized by a very high overall p_LLPS_ of 0.9887. Furthermore, this protein contains six droplet-promoting regions and seven aggregation hot-spots. The bottom plot represents the cellular context dependence of BLF3872, where the *S_BIND_* values characterize the ability of residues to switch between different binding modes. Here, regions with the context-dependent interactions change protein interaction behavior and binding modes under different cellular conditions. The corresponding residues/regions can be ordered or disordered with *S_BIND_* ≥ 2.25 (red line) [49].

**Figure 3 antibiotics-12-00471-f003:**
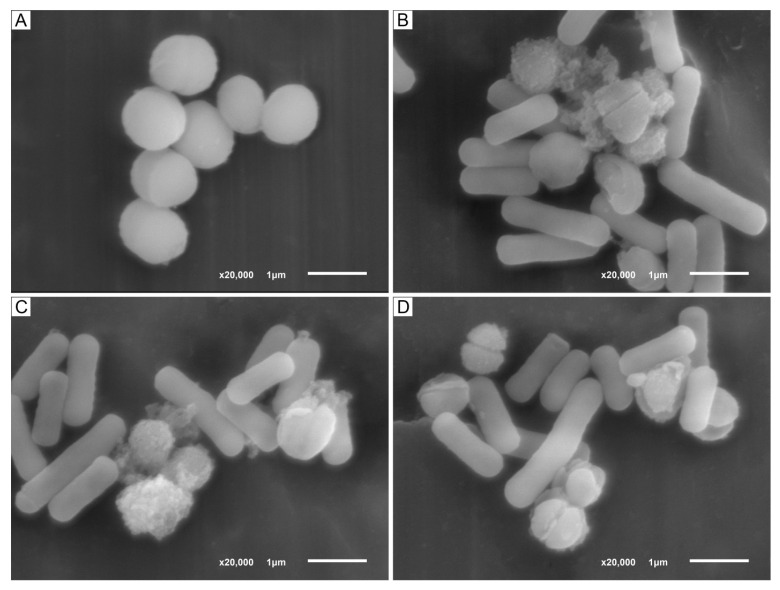
(**A**). SEM images of intact *S. aureus* 8325-4 cells. (**B**–**D**). Co-aggregation of LF3872 cells with *S. aureus* 8325-4 cells; destruction of cell wall and death of the pathogen. Note: the ratio of *S. aureus* 8325-4 cells/ LF3872 cells is 1/10.

**Table 1 antibiotics-12-00471-t001:** Antagonistic activity of LF3872 against a collection of the ATCC methicillin-resistant *S. aureus* strains.

Strain	0 h			24 h		
	C ^1^	JC ^2^	HJC ^3^	C ^1^	JC ^2^	HJC ^3^
*S. aureus* ATCC BAA-1683	3 × 10^8^	3 × 10^8^	2 × 10^8^	7 × 10^8^	<10^2^	6 × 10^8^
*S. aureus* ATCC BAA-2313	2 × 10^8^	2 × 10^8^	2 × 10^8^	6 × 10^8^	<10^2^	6 × 10^8^
*S. aureus* ATCC 33592	1 × 10^8^	2 × 10^8^	2 × 10^8^	5 × 10^8^	<10^2^	5 × 10^8^
*S. aureus* ATCC BAA-2094	4 × 10^8^	4 × 10^8^	4 × 10^8^	8 × 10^8^	<10^2^	8 × 10^8^

^1^ Control sample, amount of *S. aureus* cells in monoculture (CFU/mL). ^2^ Amount of *S. aureus* cells in co-cultivation with LF3872 (CFU/mL). ^3^ Amount of *S. aureus* cells in co-cultivation with heat-treated (56 °C, 30 min) LF3872 (CFU/mL). The results of six independent experiments in triplicates are represented.

**Table 2 antibiotics-12-00471-t002:** Antagonistic activity of LF3872 against *S. aureus* clinical isolates from the milk of women with mastitis.

Strain	0 h			24 h		
	C ^1^	JC ^2^	HJC ^3^	C ^1^	JC ^2^	HJC ^3^
*S. aureus* IIE CI-SA Hu 1246	2 × 10^8^	2 × 10^8^	3 × 10^8^	6 × 10^8^	<10^2^	7 × 10^8^
*S. aureus* IIE CI-SA Hu 1247	3 × 10^8^	3 × 10^8^	3 × 10^8^	7 × 10^8^	<10^2^	7 × 10^8^
*S. aureus* IIE CI-SA Hu 1248	8 × 10^8^	9 × 10^8^	8 × 10^8^	3 × 10^8^	<10^2^	3 × 10^8^
*S. aureus* IIE CI-SA Hu 1249	9 × 10^8^	8 × 10^8^	9 × 10^8^	4 × 10^8^	<10^2^	4 × 10^8^

^1^ Control sample, amount of *S. aureus* cells in monoculture (CFU/mL). ^2^ Amount of *S. aureus* cells in co-cultivation with LF3872 (CFU/mL). ^3^ Amount of *S. aureus* cells in co-cultivation with heat-treated (56 °C, 30 min) LF3872 (CFU/mL). The results of six independent experiments in triplicates are represented.

**Table 3 antibiotics-12-00471-t003:** Antagonistic activity of LF3872 against *S. aureus* isolated in humans from the nasopharynx and oral cavity.

Strain	0 h			24 h		
	C ^1^	JC ^2^	HJC ^3^	C ^1^	JC ^2^	HJC ^3^
*S. aureus* IIE CI-SA Hu 1261	3 × 10^8^	3 × 10^8^	3 × 10^8^	7 × 10^8^	<10^2^	7 × 10^8^
*S. aureus* IIE CI-SA Hu 1262	8 × 10^7^	9 × 10^7^	8 × 10^7^	4 × 10^8^	<10^2^	4 × 10^8^
*S. aureus* IIE CI-SA Hu 1263	4 × 10^8^	4 × 10^8^	4 × 10^8^	8 × 10^8^	<10^2^	8 × 10^8^
*S. aureus* IIE CI-SA Hu 1264	2 × 10^8^	3 × 10^8^	2 × 10^8^	7 × 10^8^	<10^2^	7 × 10^8^

^1^ Control sample, amount of *S. aureus* cells in monoculture (CFU/mL). ^2^ Amount of *S. aureus* cells in co-cultivation with LF3872 (CFU/mL). ^3^ Amount of *S. aureus* cells in co-cultivation with heat-treated (56 °C, 30 min) LF3872 (CFU/mL). The results of six independent experiments in triplicates are represented.

**Table 4 antibiotics-12-00471-t004:** Antagonistic activity of LF3872 against *S. aureus* isolated from the oropharynx of pigs.

Strain	0 h			24 h		
	C ^1^	JC ^2^	HJC ^3^	C ^1^	JC ^2^	HJC ^3^
*S. aureus* IIE CI-SA Pi 1345	8 × 10^7^	8× 10^7^	8 × 10^7^	4 × 10^8^	<10^2^	4 × 10^8^
*S. aureus* IIE CI-SA Pi 1347	2 × 10^8^	3 × 10^8^	2 × 10^8^	7 × 10^8^	<10^2^	7 × 10^8^
*S. aureus* IIE CI-SA Pi 1356	9 × 10^7^	9 × 10^7^	8 × 10^7^	4 × 10^8^	<10^2^	4 × 10^8^
*S. aureus* IIE CI-SA Pi 1357	1 × 10^8^	2 × 10^8^	1 × 10^8^	6 × 10^8^	<10^2^	6 × 10^8^

^1^ Control sample, amount of *S. aureus* cells in monoculture (CFU/mL). ^2^ Amount of *S. aureus* cells in co-cultivation with LF3872 (CFU/mL). ^3^ Amount of *S. aureus* cells in co-cultivation with heat-treated (56 °C, 30 min) LF3872 (CFU/mL). The results of six independent experiments in triplicates are represented.

**Table 5 antibiotics-12-00471-t005:** Antagonistic activity of LF3872 against *S. aureus* clinical isolates from the milk of cows with mastitis.

Strain	0 h			24 h		
	C ^1^	JC ^2^	HJC ^3^	C ^1^	JC ^2^	HJC ^3^
*S. aureus* IIE CI-SA Co 1280	8 × 10^7^	9 × 10^7^	8 × 10^7^	9 × 10^8^	<10^2^	9 × 10^8^
*S. aureus* IIE CI-SA Co 1281	2 × 10^8^	3 × 10^8^	2 × 10^8^	8 × 10^8^	<10^2^	8 × 10^8^
*S. aureus* IIE CI-SA Co 1282	1 × 10^8^	2 × 10^8^	1 × 10^8^	7 × 10^8^	<10^2^	7 × 10^8^
*S. aureus* IIE CI-SA Co 1283	3 × 10^8^	3 × 10^8^	3 × 10^8^	8 × 10^8^	<10^2^	8 × 10^8^

^1^ Control sample, amount of *S. aureus* cells in monoculture (CFU/mL). ^2^ Amount of *S. aureus* cells in co-cultivation with LF3872 (CFU/mL). ^3^ Amount of *S. aureus* cells in co-cultivation with heat-treated (56 °C, 30 min) LF3872 (CFU/mL). The results of six independent experiments in triplicates are represented.

**Table 6 antibiotics-12-00471-t006:** Co-aggregative abilities of LF3872 with antibiotic-resistant *S. aureus* strains isolated from humans and animals.

Strain	20 °C, pH 7.0 ^1^	20 °C, pH 5.0 ^2^
*S. aureus* ATCC BAA-1683	51.7 ± 4.6	87.4 ± 5.5 *
*S. aureus* ATCC BAA-2313	52.5 ± 5.7	89.3 ± 7.6 *
*S. aureus* ATCC 33592	46.8 ± 5.2	83.4 ± 6.3 *
*S. aureus* ATCC BAA-2094	45.9 ± 6.5	86.5± 5.8 *
*S. aureus* IIE CI-SA Hu 1250	54.3 ± 5.4	88.7 ± 6.6 *
*S. aureus* IIE CI-SA Hu 1247	49.4 ± 5.0	81.5 ± 5.3 *
*S. aureus* IIE CI-SA Hu 1248	48.6 ± 4.5	82.3 ± 5.9 *
*S. aureus* IIE CI-SA Hu 1249	51.7 ± 5.2	85.9 ± 6.4 *
*S. aureus* IIE CI-SA Hu 1261	49.7 ± 6.1	84.5 ± 5.3 *
*S. aureus* IIE CI-SA Hu 1262	53.6 ± 4.8	86.2 ± 5.7 *
*S. aureus* IIE CI-SA Hu 1263	54.5 ± 4.9	81.8 ± 5.4 *
*S. aureus* IIE CI-SA Hu 1264	48.3 ± 4.5	80.6 ± 5.8 *
*S. aureus* IIE CI-SA Pi 1345	52.9 ± 3.8	87.4 ± 5.9 *
*S. aureus* IIE CI-SA Pi 1347	54.5 ± 4.7	86.5 ± 5.5 *
*S. aureus* IIE CI-SA Pi 1348	49.6 ± 4.5	83.8 ± 5.7 *
*S. aureus* IIE CI-SA Pi 1356	53.7 ± 4.9	85.4 ± 5.4 *
*S. aureus* IIE CI-SA Co 1280	51.4 ± 4.6	86.3 ± 5.8 *
*S. aureus* IIE CI-SA Co 1281	48.7 ± 5.0	83.7 ± 4.9 *
*S. aureus* IIE CI-SA Co 1282	54.2 ± 5.3	87.5 ± 6.5 *
*S. aureus* IIE CI-SA Co 1283	53.8 ± 4.7	84.7 ± 6.2 *

^1,2^-Co-aggregation with LF3872 (%). The percentage of co-aggregation was measured in PBS buffer at pH 7.0 and in MES buffer at pH 5.0. * *p* < 0.05 co-aggregation of LF3872 with *S. aureus* strains at pH 5.0 vs. pH 7.0. The results of six independent experiments in triplicates are represented.

**Table 7 antibiotics-12-00471-t007:** Extracellular ATP levels in antibiotic-resistant *S. aureus* strains treated with CSLF3872.

Strain	Boiled CSLF3872 ^1^	CSLF3872 ^2^
*S. aureus* ATCC BAA-1683	5.8 ± 1.6	342.4 ± 15.8 ***
*S. aureus* ATCC BAA-2313	5.5 ± 1.4	358.2 ± 14.6 ***
*S. aureus* ATCC 33592	4.9 ± 1.7	320.8 ± 14.6 ***
*S. aureus* ATCC BAA-2094	5.6 ± 1.8	325.9 ± 12.3 ***
*S. aureus* IIE CI-SA Hu 1250	4.7 ± 1.5	336.5 ± 15.4 ***
*S. aureus* IIE CI-SA Hu 1247	6.3 ± 2.1	325.6 ± 12.8 ***
*S. aureus* IIE CI-SA Hu 1248	7.5 ± 1.4	365.5 ± 18.2 ***
*S. aureus* IIE CI-SA Hu 1249	8.6 ± 2.2	370.6 ± 16.5 ***
*S. aureus* IIE CI-SA Hu 1261	6.3 ± 2.1	334.7 ± 14.6 ***
*S. aureus* IIE CI-SA Hu 1262	7.6 ± 1.8	325.3 ± 15.4 ***
*S. aureus* IIE CI-SA Hu 1263	5.0 ± 1.7	340.4 ± 13.5 ***
*S. aureus* IIE CI-SA Hu 1264	7.4 ± 1.8	356.5 ± 21.0 ***
*S. aureus* IIE CI-SA Pi 1345	6.3 ± 1.5	327.3 ± 18.5 ***
*S. aureus* IIE CI-SA Pi 1347	8.5 ± 2.3	335.1 ± 16.7 ***
*S. aureus* IIE CI-SA Pi 1348	6.4 ± 1.5	354.7 ± 19.6 ***
*S. aureus* IIE CI-SA Pi 1356	7.9 ± 2.4	347.3 ± 16.4 ***
*S. aureus* IIE CI-SA Co 1280	5.8 ± 1.3	355.6 ± 18.6 ***
*S. aureus* IIE CI-SA Co 1281	7.2 ± 2.5	364.3 ± 20.5 ***
*S. aureus* IIE CI-SA Co 1282	6.9 ± 1.7	347.7 ± 18.3 ***
*S. aureus* IIE CI-SA Co 1283	7.5 ± 1.8	365.4 ± 21.8 ***

^1,2^ Concentration of ATP (nm/OD). *** *p* < 0.001 levels of extracellular ATP in strains of *S. aureus* (control, boiled CSLF3872) vs. native CSLF3872. The results of six independent experiments in triplicates are represented.

**Table 8 antibiotics-12-00471-t008:** Dynamics of lactic production in the process of LF3872 strain cultivation in the MRS broth.

**Cultivation Time, h**	5	10	15	20
**Lactic Acid Production, µg/mL**	97 ± 4	164 ± 6	207 ± 5	210 ± 8

The results of six independent experiments in triplicates are represented.

**Table 9 antibiotics-12-00471-t009:** Hydrogen peroxide in the process of LF3872 strain cultivation in the MRS broth.

**Cultivation Time, h**	5	10	15	20
**H_2_O_2_ Concentration, mM/L in CS of LF3872**	0.83 ± 0.2	0.84 ± 0.3	0.92 ± 0.5	0.85 ± 0.4

The results of six independent experiments in triplicates are represented.

**Table 10 antibiotics-12-00471-t010:** Microorganisms used in this study.

Microorganism	Strain	Antibiotic Resistance of *S. aureus*	Growth Conditions
*L. fermentum*	IIE ^1^ 3872		MRS ^a^ 37 °C anaerobically 48 h
*S. aureus*	8325-4 ^2^	GEN	TGVC ^b^ 37 °C aerobically 24 h
*S. aureus*	ATCC ^3^ BAA-1683	MET	The same
*S. aureus*	ATCC BAA-2313	MET	The same
*S. aureus*	ATCC 33592	MET, GEN	The same
*S. aureus*	ATCC BAA-2094	MET, OXC, BENPEN	The same
*S. aureus*	IIE CI-SA Hu 1250 ^4^	TET, ERY, NAL, CEFA	The same
*S. aureus*	IIE CI-SA Hu 1247 ^5^	AMP, OXA, PEN, TET	The same
*S. aureus*	IIE CI-SA Hu 1248 ^6^	TET, NAL, CIP, ERY	The same
*S. aureus*	IIE CI-SA Hu 1249 ^7^	GEN, CIP, VAN, METR	The same
*S. aureus*	IIE CI-SA Hu 1261 ^8^	PEN, OXA, VAN, CEFA	The same
*S. aureus*	IIE CI-SA Hu 1262 ^9^	PEN, VAN, METR, CIP	The same
*S. aureus*	IIE CI-SA Hu 1263 ^10^	PEN, VAN, METR, ERY	The same
*S. aureus*	IIE CI-SA Hu 1264 ^11^	PEN, VAN, TET, MET	The same
*S. aureus*	IIE CI-SA Pi 1345 ^12^	MET, OXA, PEN, BENPEN, CEFO	The same
*S. aureus*	IIE CI-SA Pi 1347 ^13^	MET, PEN, CEFA, CEFO	The same
*S. aureus*	IIE CI-SA Pi 1348 ^14^	MET, PEN, OXA, CEFA, CEFO	The same
*S. aureus*	IIE CI-SA Pi 1356 ^15^	MET, PEN, OXA, ERY, CEFA, CEFO	The same
*S. aureus*	IIE CI-SA Co 1280 ^16^	NAL, GEN, ERY, CEFA	The same
*S. aureus*	IIE CI-SA Co 1281 ^17^	CIP, PEN, CEFA, METR	The same
*S. aureus*	IIE CI-SA Co 1282 ^18^	CIP, PEN, VAN, METR	The same
*S. aureus*	IIE CI-SA Co 1283 ^19^	CIP, PEN, CEFO, METR	The same

^1^ Collection of Microorganisms at the Institute of Immunological Engineering (IIE), Department of Biochemistry of Immunity and Biodefence, Lyubuchany, Moscow Region, Russia. ^2^ Collection of Microorganisms at the National Research Center of Epidemiology and Microbiology named after Academician N.F. Gamaleya of the Ministry of Health of the Russian Federation. ^3^ American Type Culture Collection, Manassas, VA, USA. ^4–7^ Clinical isolates from the milk of women with mastitis. ^8–11^ Clinical isolates obtained from the human nasopharynx and oral cavity. ^12–15^ Clinical isolates of MRSA strains from the oropharynx of pigs. ^16–19^ Clinical isolates obtained from the milk of cows with mastitis (CM). ^a^ Man-Rogosa-Sharp (MRS) broth or agar-containing plates (HiMedia, India). ^b^ TGVC broth or agar (HiMedia, India). Antibiotic resistance: GEN—gentamicin, MET—methicillin, TET—tetracycline, AMP—ampicillin, PEN—penicillin, NAL—nalidixic acid, CIP—ciprofloxacin, OXC—oxacillin, BENPEN—benzylpenicillin, CEFA—cefalexin, CEFO-cefotaxime, VAN—vancomycin, METR-metronidazole.

## Data Availability

Not applicable.

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
