# Peer review of "Limosilactobacillus fermentum 3872 That Produces Class III Bacteriocin Forms Co-Aggregates with the Antibiotic-Resistant Staphylococcus aureus Strains and Induces Their Lethal Damage"

_antibiotics, 2023, doi:10.3390/antibiotics12030471_

Round 1
Reviewer 1 Report
Minor comments:
Abstract is wordy and should be more concise.
Line 95-102: It is not relevant to this study. Please consider to make it more concise.
Line 120 to 228: Results and Discussion should come first before Materials and Methods. Please refer to Antibiotics journal format.
Line 139 to 180 can be more concise.
Line 192 to 207 are explained in simple way. I understand the authors cited other papers, but please add brief explanation to the methods.
Table 3: What is JC, C, and HJC? Please label the table appropriately.
Similarly all tables should be labeled appropriately. I noticed some legends are missing.
Conclusion is too long and should be more concise.
In summary I believe the data presented in this manuscript warrant publication in Antibiotics. But the manuscript is too long and not organized. The authors should try to revise the manuscript to make it more concise.
Reviewer 2 Report
The manuscript is interesting but needs to be improved for most readers of the journal. The revision should be performed especially the linkage with the clinical application for many practitioners.
For the general comments, the introduction part should introduce more on a new class III bacteriocin BLF3872. What is already found on this BLF3872? I also would like to know how important Sequence alignment and bioinformatic analysis, scanning electron microscopy, lactic acid determination, and hydrogen peroxide determination findings are.
I would suggest merging the first 4 paragraphs together and explaining more about the bactericidal action of antibiotics in various S. aureus strains.
The authors need to explain 2.4 “Determination of antibacterial activity of LF3872 against the studied groups of S. aureus 192 strain” and 2.5 “Co-aggregation assay for determination of interactions between LF3872 and S. aureus strains” in detail because their results were important.
Results on the antibacterial activity of LF3872 (3.3-3.7) and Table 2-6 should be merged together for easy comparisons.
Please discuss more the findings on Cells of LF3872 co-aggregate with antibiotic-resistant S. aureus strains (3.8) and CSLF3872 induces cell damage and ATP leakage (3.9).
For the discussion of 3.10 and 3.11, the authors measured lactic acid and H2O2 produced by LF3872 but no other groups for comparison in the study. Therefore, discussion on the over studies may not be a good reference. Please make a revision.
For the discussion on Hydrogen peroxide production, the authors compared the results of LF3872 with L. gasseri. Since there is no relationship between LF3872 and L. gasseri, the authors need to clarify the reason for the insignificant of H2O2 among times.
Reviewer 3 Report
There are some redundances, check the whole text
Specify better : resistance of strain to resistant S aureus
